# Development of a self-management intervention for stroke survivors with aphasia using co-production and behaviour change theory: An outline of methods and processes

**Faye Wray**[1,2]*, **David Clarke**[1,2], **Madeline Cruice**[3], **Anne Forster**[1,2]

**1** Academic Unit for Ageing and Stroke Research, Leeds Institute of Health Sciences, University of Leeds, Leeds, United Kingdom, **2** Academic Unit for Ageing and Stroke Research, Bradford Teaching Hospitals NHS Foundation Trust, Bradford Institute for Health Research, Bradford, United Kingdom, **3** School of Health Sciences, City University of London, London, United Kingdom

* f.d.wray@leeds.ac.uk

## Abstract

### Background

Self-management is a promising approach to improve quality of life after stroke. However, evidence for the appropriateness and effectiveness of self-management for stroke survivors with aphasia is limited. This article reports on the process used to develop a supported self-management intervention for stroke survivors with aphasia (SSWA) using co-production and behaviour change theory. Preparatory research included systematic reviews, and qualitative interviews and focus groups with SSWA, family members and speech and language therapists (SLTs).

### Materials and methods

We conducted six, 2 hour long intervention development workshops with key stakeholders. The workshops were informed by principles of co-production and the intervention development process outlined by the Behaviour Change Wheel (BCW). We also incorporated the findings of our preparatory research within workshops. Each workshop included an introduction, 1–2 co-production tasks and time for feedback at the end of the session. Data were analysed on an ongoing basis so that findings could be used to feed in to subsequent workshops and intervention development.

### Results

Workshop participants (n = 12) included; SSWA (n = 5), family members (n = 3) and SLTs (n = 4). Together, participants engaged with accessible and participatory co-production tasks which aligned with the BCW framework. Participants engaged in discussion to define self-management in behavioural terms (behavioural diagnosis) and to identify what needed to change to support self-management. Participant's co-produced solutions for supporting self-management and discussed options to implement these in practice. Prototype materials

**Data Availability Statement:** All relevant data are within the manuscript.

**Funding:** Funding Funding for this study was provided by a postdoctoral fellowship awarded to FW by The Stroke Association (Ref: SA PDF 19 \100011) (https://www.stroke.org.uk/research/support-programme-aphasia-life-after-stroke). Authors DC, MC and AF were co-applicants on this fellowship. The funder did not play a role in the design of the study, or in the collection and analysis of data, or in writing the manuscript, or in the decision to publish.

**Competing interests:** The authors have declared that no competing interests exist.

were generated by the research team and evaluated by participants. Intervention functions and behaviour change techniques (BCTs) were mapped to the solutions generated by participants by the research team, after the final workshop. A supported self-management intervention for SSWA was developed which will be delivered by SLTs through community stroke services.

## Conclusions

This paper reports the process we used to integrate co-production work with behaviour change theory to develop a complex self-management intervention. This is of relevance for researchers looking to harness the strengths of co-production methods and theory in intervention design. Future research will feasibility test the supported self-management intervention developed. This paper provides transparency to our intervention development process which will help others to better interpret the findings of our feasibility work.

## Background

Stroke remains a leading cause of death and disability worldwide [1]. Approximately one third of stroke survivors will experience aphasia; an impairment affecting the comprehension of expression of language across one or more modalities (spoken, written or sign language) [2]. Stroke survivors with aphasia (SSWA) experience particularly poor longer-term outcomes including reduced quality of life [3], reduced social participation [4] and, an increased risk of depression [5]. Qualitative studies also highlight the persistent difficulties faced by this population in adjusting to and managing life with aphasia [6]. This is not limited to the acute period; those who are several years post-stroke also express ongoing difficulties in maintaining social networks and participating in valued activities [6].

Efforts to develop an evidence-based pathway for longer-term care in stroke have increased in recent years [7–10]. One promising approach to improve longer-term outcomes is 'self-management'. Self-management interventions aim to empower patients with the knowledge and skills that they need to adjust to and manage the consequences of living with a long-term condition [11–13]. Such interventions typically include multiple components (e.g. education, goal setting, action planning and decision making) and have been delivered in various formats (e.g. group-based, telephone based, individually tailored) [11–13].

Self-management is recommended as part of longer-term stroke care in a number of countries including the UK [14, 15], Australia [16], USA [17] and Canada [18]. A Cochrane review suggested such interventions may improve stroke survivors quality of life and increase self-efficacy [12]. However, this evidence base is limited in three ways. Firstly, SSWA are underrepresented in this research and therefore, the effectiveness of self-management approaches for this group is unclear [19]. Secondly, given that typical components of self-management (e.g. education, action planning, decision making) are mediated through language, the accessibility of such approaches for SSWA is questionable [19]. Thirdly, such approaches may not contain components which are designed specifically to address the unique barriers to life participation posed by aphasia [6].

We set out to develop a self-management intervention which was accessible to SSWA and which met the specific needs of this population in terms of managing the day-to-day consequences of living with a communication difficulty.

Methods for developing complex interventions are evolving rapidly, however, there is a lack of evidence about which methods might lead to the most effective interventions [20].

Numerous approaches have been used and these have recently been summarised and categorised by O'Cathain et al. [21]. They include: 1) Partnership, 2) Target-population centred, 3) Evidence and theory-based, 4) Implementation-based, 5) Efficiency-based 6) Stepped or phased approaches 7) Intervention-specific, and 8) Combination approaches. Such a taxonomy is useful for providing a guiding framework of approaches. Recent guidance has also emphasised the iterative nature of complex intervention development and the need for flexibility in adapting methods to suit different contexts or populations [20].

Innovative intervention development processes which incorporate the use of evidence, theory or stakeholder involvement have begun to be reported in the development of self-management interventions in other long-term conditions [22–24]. Transparent reporting of the intervention development process is key to advancing our understanding of which intervention development approaches are most effective [20]. This is particularly relevant in stroke where the intervention development process is often poorly reported [25]. This may produce research waste as a lack of rigorous development process may give the intervention less chance of being effective and may also negatively impact on translation to real world settings [20].

The aim of this paper is to report the methods and processes we used to develop a self-management intervention for SSWA. To develop the intervention we used a partnership (co-production) [26, 27] approach in combination with established behaviour change theory [28, 29]. Theory provides a useful basis to underpin the design of the 'active' components of an intervention (those thought to produce behaviour change) [28, 30]. It is also a useful basis for evaluating the intervention; as pre-defined components can be assessed for effectiveness and to understand how the intervention may need to be refined [31]. Partnership approaches involve key stakeholders working together with the research team to develop an intervention [21]. The proposed benefits are that the design of the intervention is driven by the requirements of those who will use it; increasing the likelihood that the intervention will be acceptable and feasible to deliver in practice [32, 33]. Established guidance from the Medical Research Council (MRC) recommends incorporating stakeholder involvement and the use of theory within the process of complex intervention development but lacks detail about how this may be done in practice [34].

We wished to harness the potential benefits of a theory-based and partnership-based approach. In this paper we outline how we combined these approaches in the development of the intervention. Reporting transparently in this way will aid other researchers in developing complex interventions and will provide appropriate context to interpret the results of planned feasibility testing.

### Aim

To report the methods and processes used to develop a self-management intervention for SSWA using a theory-based and partnership-based approach.

## Materials and methods

### Preparatory research

Preparatory research to inform development of the intervention included: 1) Systematic reviews of the literature, and 2) A qualitative needs assessment. We systematically reviewed randomised controlled trials (RCTs) of existing self-management interventions in stroke [19]. We also systematically reviewed qualitative literature relating to longer-term needs of stroke survivors with communication difficulties [6]. We undertook qualitative interviews and focus groups with SSWA (n = 15), their family members (n = 10) and speech and language therapists (SLTs) (n = 18) [35, 36] to understand their needs and priorities in relation to self-

management support. We detail in Table 1 how the preparatory research fed in to the development of the intervention.

## Overview of study design

We designed this study with reference to the methods used by Hall et al. [37]. We ran a series of six intervention development workshops with key stakeholders (SSWA, their family members and SLTs), over a seven month period, to develop the self-management intervention. Ethical approval for this study was granted in September 2019 by North West- Preston Research Ethics Committee (19/NW/0531). A summary of intervention development process and methods is shown in Fig 1.

The intervention development workshops were underpinned by principles of co-production [26, 27]. Terminology surrounding partnership approaches (such as co-production or co-design) is often used interchangeably [26] and has been criticised for being poorly defined [38]. In this study, we defined co-production as a collaborative process in which key stakeholders (SSWA, their family members and SLTs) worked together in a structured and facilitated way [26, 27]. In this process, we assumed equality between stakeholders; who each brought important knowledge and expertise to the co-production process [39]. We also recognised co-production to be a cyclical and iterative process where feedback from one meeting fed in to the next [40].

The content of the intervention development workshops was guided by the Behaviour Change Wheel (BCW) [28, 29]; a framework for designing theory based behaviour change interventions. The BCW is based on an underpinning theory of behaviour change; the COM-B model which hypothesises that capability (C), opportunity (O) and motivation (M) are needed in order for a behaviour to occur [28, 29]. The 'active' ingredients of the intervention are the Behaviour Change Techniques (BCTs) which work to change C, O, or M. The stages of designing an intervention using the BCW include:

- **Stage One: Understand the behaviour** (including: a. defining the problem in behavioural terms, b. selecting the target behaviour, c. specifying the target behaviour, d. identifying what needs to change to achieve the target behaviour)

- **Stage Two: Identifying intervention options** (including: a. identifying intervention functions, b. identifying policy categories).

- **Stage Three: Identifying intervention content and implementation options** (including: a. linking intervention functions with BCTs and b. identifying the mode of delivery).

We provide further detail about how the BCW was used the structure the intervention development workshops in the sections which follow. Some stages of the BCW were not undertaken in the intervention development workshops and were undertaken by members of the research team prior to the first workshop, between workshops or after the final workshop. For example, for stage 1b, the research team identified five candidate behaviours prior to the first workshop (based on the preparatory research) which were subsequently refined based on discussions in workshop one and two. For stage 2b (identifying policy categories), many of the policy categories were not applicable to the current intervention so we prioritised tasks relating to other aspects of the BCW to maximise productivity within the limited number of workshops available. For stage 3a (linking intervention functions with BCTs) we felt it would not be possible to make this task accessible to participants (in particular SSWA) and so it was completed by the research team after the final workshop.

**Table 1. Overview of workshop content.**

| Title of workshop and summary of content | Co-production tasks | Supporting materials | Links to BCW[a] stages (sub-stages) | Integration of evidence from preparatory research |
|---|---|---|---|---|
| **1. Preparatory meeting**<br>• Introduction to the topic of self-management and key behaviours including supporting videos [44, 45].<br>• Introduction to co-production and how we expected the groups would run.<br>• Mini-co-production task: What does living well with aphasia mean to you? | 1. Ice-breaker: Participants asked to bring in a personal object to help introduce themselves.<br>2. Discussion: what does living well with aphasia mean to you? | • Self-management infographic<br>• Co-production infographic | • Stage one: Understand the behaviour (1a Define the problem in behavioural terms, 1b Select target behaviour) | Presentation on self-management and self-management infographic informed by systematic reviews [6, 19] and qualitative interviews with SSWA[b] [35]. |
| **2. Talking about 'self-management'?**<br>• Video presentation of a SSWA discussing how aphasia impacted their day to day life [46].<br>• Focus on what the term 'self-management' means for SSWA including challenges and opportunities. | 1. Discussion of challenges SSWA in video may face and what self-management may mean to them.<br>2. Discussion of the term 'self-management'. Is this term liked? What other words could we use to describe self-management? | • Summary of video in written form<br>• Challenges worksheet<br>• Communicating about self-management worksheet | • Stage one: Understand the behaviour (1c Specify the target behaviour) | Video aligned with key themes arising from qualitative interviews with SSWA [35]. |
| **3. What can we change to support self-management?**<br>• Participants were asked to generate solutions for supporting self-management (including considering barriers to key behaviours and who might help and when).<br>• Discussions were illustrated by an artist in real time and were pinned around the room for participants to look at and comment on. | 1. What might help in supporting; communication outside of the home, seeing family and friends, taking the lead in managing and taking part in enjoyable activities. Barriers to each behaviour were presented with space for participants to add their own suggestions.<br>2. Who might be involved in supporting self-management and when. | • What may help support self-management worksheet<br>• Who might help support self-management and when | • Stage one: Understand the behaviour (1d Identify what needs to change)<br>• Stage two: Identify intervention options (2a Intervention functions) | Barriers to key behaviours drawn from qualitative interviews with SSWA and SLTs [35, 36] and systematic review of qualitative research [6] |
| **Title of workshop and summary of content** | **Co-production tasks** | **Supporting materials** | **Links to BCW[a] stages (sub-stages)** | **Integration of evidence from preliminary research** |
| **4. Which solutions should we use to support self-management?**<br>• Participants prioritised the solutions they had generated in the last workshop.<br>• Participants reviewed existing written resources; commenting on their pros and cons.<br>• The artist presented examples of illustrations which could be used to support provision of information. | 1. Discussion of most important, least important and any 'missing' solutions<br>2. 'Speed dating' of existing written resources. Discussion of pros and cons including formatting. | • Summary of solutions infographic<br>• Feedback on solutions worksheet<br>• Feedback on existing written resources worksheet | • Stage two: Identify intervention options (2a Intervention functions)<br>• Stage three: Identify content and implementation options (3b Mode of delivery) | |
| **5. Prototype solutions for supporting self-management**<br>• The research team presented a prototype solution for the intervention.<br>• Participants reviewed prototype materials and considered how we could encourage people to use each component of the intervention in practice. | 1. Participants reviewed prototype materials for the intervention. Facilitators/participants wrote suggested changes directly on to the prototypes. | • Summary of intervention components<br>• Prototype materials | • Stage two: Identify intervention options (2a Intervention functions)<br>• Stage three: Identify content and implementation options (3b Mode of delivery) | |

*(Continued)*

**Table 1.** (Continued)

| Title of workshop and summary of content | Co-production tasks | Supporting materials | Links to BCW<sup>a</sup> stages (sub-stages) | Integration of evidence from preparatory research |
|---|---|---|---|---|
| **6. How can we put the self-management approach we have developed in to practice?**<br>• Participants reviewed prototype materials.<br>• Evaluation of the workshops. | 1. Prototype materials were shared on screen and the facilitator took notes as participants commented on these.<br>2. Evaluative questions were shared on screen and the facilitator took notes as participants commented on these. | • Prototype materials<br>• Prototype materials worksheet<br>• Questions to guide evaluation discussion | • Stage two: Identify intervention options (2a Intervention functions)<br>• Stage three: Identify content and implementation options (3b Mode of delivery) | |

Abbreviations:

[a] BCW- Behaviour Change Wheel;

[b] SSWA- Stroke survivors with aphasia

### Intervention development groups

Intervention development workshops took place in private room, in a community venue owned by a not-for-profit organisation. The groups were facilitated by the first author and one other researcher. Both facilitators were experienced researchers, were female, educated to PhD level and had a disciplinary background in Psychology. The groups lasted for approximately 2 hours. Although all groups were planned to be held face-to-face, due to COVID-19 the last workshop was held by videoconference. To maximise accessibility, we undertook the final workshop in smaller groups with SSWA and their family members participating in a separate videocall to SLTs.

**Participants.** We aimed to recruit 3–4 participants from three key stakeholder groups: 1) SSWA 2) the family members or friends of SSWA and 3) SLTs (12 participants in total). SSWA and their family members were a convenience sample, recruited from local stroke groups and through their involvement in the preparatory research phases. SSWA were eligible to participate if they were aged 16 or over, had post-stroke aphasia (as diagnosed by the treating speech and language therapy service or as self-reported by the stroke survivor), were able and willing to provide informed consent, were English-speaking and were able to attend the dates of at least four of the six workshop sessions. No restrictions were made in terms of time post-stroke. Family members or close friends were eligible to participate if they were aged 16 or over, were a family member or close friend of a person with aphasia participating in the study, were English speaking and were able to attend at least four of the workshop sessions. SLTs were experienced clinicians (UK National Health Service band 6 or above whereby a roles banding is broadly indicative of experience level e.g. band 5 = newly qualified therapist, band 6 = specialist therapist, Band 7 = highly specialist therapist/manager) who had a caseload including SSWA and were purposively sampled to represent a mixture of acute and community stroke services. SLTs were also asked to commit to attending at least four of the six workshop sessions. We did not collect data on the number of stakeholders approached to participate in the study as recruitment often occurred through gatekeepers e.g. service managers, stroke group co-ordinators and we did not wish to add any additional burden to these contacts (which may have reduced the likelihood of participation).

All participants took part in a recruitment visit with the first author prior to the intervention development workshops where informed consent was provided. We also collected demographic data at this visit. For SSWA this included the Frenchay Aphasia Screening Test (FAST) [41], a brief language assessment covering; comprehension, expression, reading and writing (possible scores ranging from 0–30 and a score of <27 being indicative of aphasia in

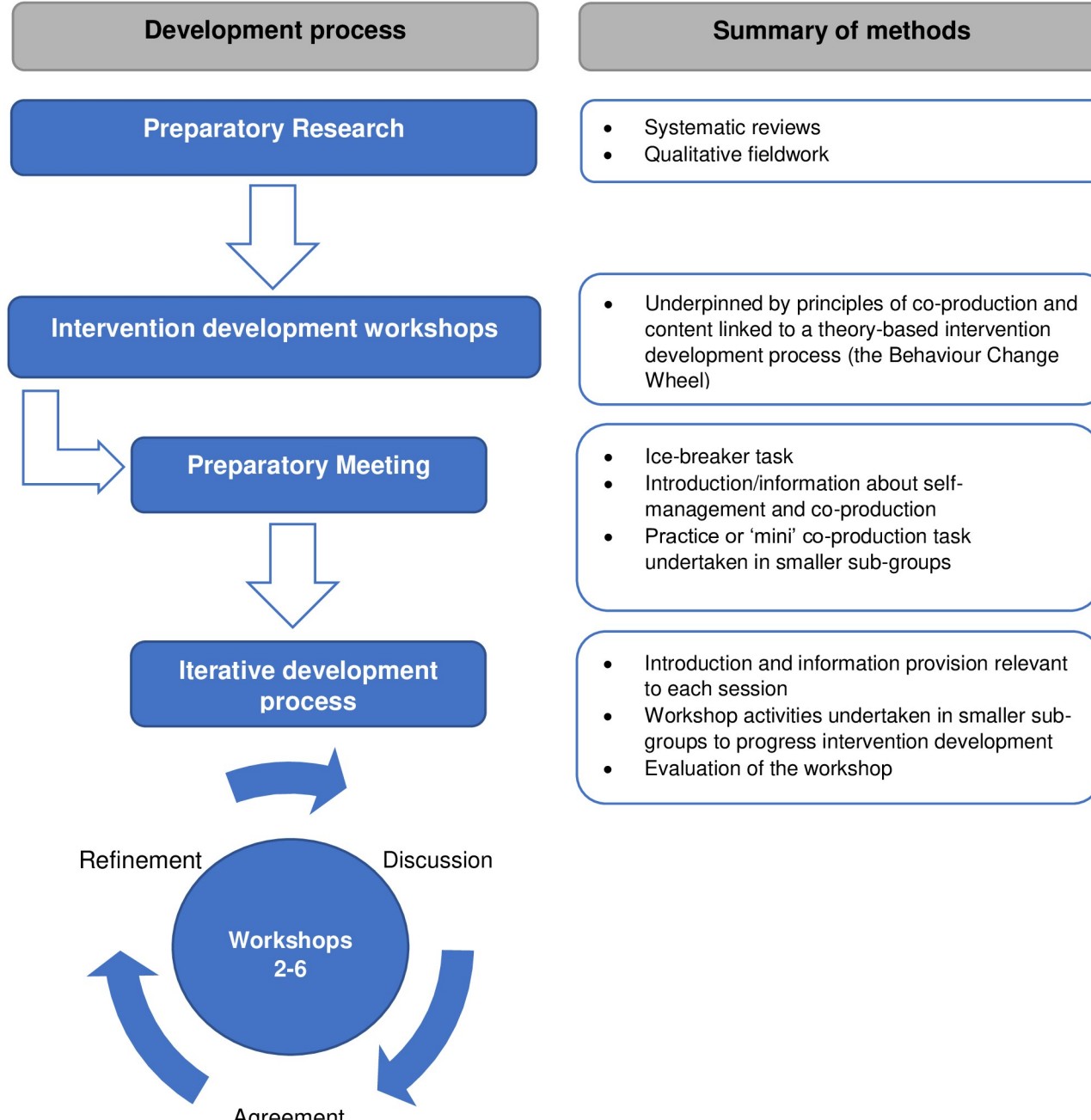

**Fig 1. Summary of intervention development process.**

those aged up to 60 and <25 in those aged 61 or over). We also included a measure of perceived everyday communication ability (Communication Outcomes after Stroke [COAST] scale) [42]. The COAST is a 20 item measure with raw scores (ranging from 0–80) expressed as a percentage of the maximum score of 80. The COAST includes items relating to perception of communication effectiveness (e.g. being able to speak to people the participant knows well, read, write) and items relating to the impact of this on their life (e.g. confidence, family life, social life).

**Preparatory meeting.** The first workshop aimed to introduce participants to the project, the research team and each other. As an ice-breaker task, we asked participants to bring in a personal object and to introduce themselves and their object at the beginning of the meeting. Following this, we used brief PowerPoint presentations to introduce the key concepts of self-management and co-production to the group. The presentations contained short video clips (which were obtained from the internet) to illustrate key messages. We also produced accessible infographics for participants about self-management and co-production. The final part of the preparatory meeting involved a 'mini co-production task' to introduce participants to the sorts of discussions they may have in subsequent workshops. In smaller groups of 3–4 participants (including a mix of SSWA, family members and SLTs) we asked participants to discuss what 'living well with aphasia' meant to them. Key words and phrases were written on a large sheet of paper by participants and/or facilitators. At the end of the group, participants rejoined in to a bigger group and fed back their comments on what they had discussed and the overall format of the meeting.

**Content of workshops.** The five workshops which followed the preparatory meeting included three main components;

1. Introduction and information provision
   During the first part of the meeting, we provided a recap of the last session and provided key information which was relevant to the current session (lasting 15-20mins). Accessible written summaries were also provided to illustrate main points.

2. Workshop activities undertaken in smaller sub-groups
   One to two, structured workshop activities were undertaken in each group (lasting 35-45mins each). Activities were broadly aligned to the three stages of the BCW framework for intervention development [28, 29] (see Table 1). Activities were designed to draw upon the expert knowledge and experience of participants to inform each stage of intervention development. However, care was taken to avoid technical language associated with this framework to maximise accessibility for participants. For example, we used the term 'problem' rather than 'behavioural diagnosis'. We also asked participants to generate their own intervention 'solutions' rather than providing lists of 'intervention functions' or 'BCTs'. The first author mapped the data generated by the activities back to the BCW framework between groups. Activities were undertaken in smaller sub-groups of 3–4 participants. Some sub-groups contained a mix of SSWA, family members and SLTs and some activities were undertaken with SSWA, family members and SLTs in separate sub-groups. Workshop activities were supported by accessible worksheets (S1 File) and infographics (S2 File) created bespoke for each task. In workshops five and six, a prototype intervention was also presented to participants. Some elements of the prototype intervention were descriptive e.g. summarising key content of intervention components and some elements were prototypes of materials to be used within the intervention such as written materials or tools.

3. Evaluation of the workshop.
   At the end of each meeting, the sub-groups joined together to evaluate the workshop. Participants gave short reflections on what had been discussed in the group. We also asked participants to say what they liked about the workshop and if there was anything they would change. Feedback from participants was used to shape subsequent workshop sessions.

**Integration of evidence from preparatory research.** Key findings from the preparatory research were integrated in the first three co-production workshops. This allowed the groups decisions to be informed by evidence where this was available. For example, in our preparatory research we had identified key behaviours which may underpin self-management and

potential barriers to these. We developed an accessible worksheet where we outlined a behaviour and the previously identified barriers ('things that might stop me') to inform participant generated solutions.

**Accessibility of workshops for stroke survivors with aphasia.** To maximise the accessibility of workshops for SSWA we used a number of strategies. Firstly, we ensured that any written information (including information provided in PowerPoint slides) was accessible by formatting this in line with Stroke Association guidance [43]. We also chose multimedia (e.g. video clips) [44–46] based on accessibility, for example, ensuring that verbal information was appropriately paced and, in most cases, that videos included subtitles to aid comprehension [45, 46]. Secondly, to maximise the accessibility of the workshop activities, we ensured that one facilitator was assigned to support SSWA when working in separate sub-groups. Both researchers who facilitated the groups have had training in supporting SSWA to communicate and have experience of conducting research groups with SSWA. Supported conversation techniques (e.g. speaking in short sentences using high-frequency words, using repetition to aid comprehension and paraphrasing responses to check understanding) were used by facilitators as required [47, 48]. Based on previous research suggesting that some family members of people with aphasia may engage in 'speaking for' behaviours [49, 50], we also took the decision to separate family members from their relative with aphasia when working in smaller sub-groups. Using this approach, we hoped to maximise the opportunity for SSWA to contribute to the discussion. Finally, in two workshops, we used a local artist to illustrate ideas and discussion in two of the workshop sessions (S3 File). This supported SSWA to communicate their ideas and acted as a point of reference to guide discussion.

## Data collection and analysis

All worksheets generated as part of the groups were utilised as data. The first author wrote notes during and after each session to capture any discussion of key decisions made (sessions were not audio recorded). The first author and the second facilitator also debriefed after each session to discuss key points made by participants. The approach to the content of the workshops (and intervention development) was iterative. After each workshop, the first author created a summary of the data collected which was further refined into an accessible summary to present to participants (S4 File). Data collected were also checked against the suggested steps for intervention development in the BCW and data mapped accordingly to intervention functions and BCTs [28, 29]. The first author undertook behaviour change taxonomy training to aid this process [51]. Based on the data collected and discussions with the wider research team, the content for the subsequent workshop was revised accordingly.

The first author took notes on the evaluation at the end of workshops 1–5. These were not analysed in any formal manner but instead used to refine how the workshops ran subsequently. Discussions from the overall evaluation in workshop 6 are summarised in the Results section based upon fieldnotes taken by the first author. The fieldnotes were not subject to any formal analysis. They were reviewed by the first author and are summarised in the Results section.

## Results

### Participants

A total of 12 stakeholders expressed an interest in the study and all went on to provide informed consent and participate in the intervention development groups. This included SSWA (n = 5), family members (n = 3) and SLTs (n = 4). Table 2 shows an overview of participant characteristics. The mean FAST score for SSWA who participated was 24.4 (range 22–26) and the mean COAST score was 47.91% (range 18.75%-68.75%).

**Table 2. Participant characteristics.**

| Stroke survivors with aphasia | | | | | Family members | | | | | Speech and language therapists[a] | | |
|---|---|---|---|---|---|---|---|---|---|---|---|---|
| Participant number | Age | Gender | Occupational status | Time post-stroke (years) | Participant number | Age | Gender | Occupational status | Relationship to SSWA | Participant number | Type of service[1] | National Health Service (NHS) Band[b] |
| 01 | 43 | Female | Unemployed | 11 | 06 | 69 | Male | Retired | Father | 09 | Hospital inpatient | 7 |
| 02 | 57 | Male | Retired | 6.5 | 07 | 63 | Male | Retired | Husband | 10 | Community stroke team | 6 |
| 03 | 56 | Female | Retired | 6 | 08 | 51 | Female | Carer | Wife | 11 | Community speech and language therapy service | 6/7 |
| 04 | 62 | Female | Retired | 14 | | | | | | 12 | Community speech and language therapy service | 6 |
| 05 | 51 | Male | Retired | 4 | | | | | | | | |

[a] Speech and language therapists were recruited from three different NHS trusts in the North of England;

[b] NHS bandings typically denote the following levels of experience: Band 6 specialist therapist, Band 7 highly specialist therapist/manager

## Attendance

Table 3 shows an overview of attendance at each group. One SSWA/family member dyad was unable to attend two of the sessions due to illness and a prior commitment. A different dyad was unable to attend one session due to a prior commitment. One SLT was unable to attend

**Table 3. Overview of group attendance.**

| Workshop | Participants (n attending) |
|---|---|
| 1. Preparatory meeting | SSWA[a] (n = 4) |
| | Family members (n = 2) |
| | SLTs[b] (n = 3) |
| 2. Talking about self-management | SSWA (n = 5) |
| | Family members (n = 3) |
| | SLTs (n = 4) |
| 3. What can we change to support self-management? | SSWA (n = 5) |
| | Family members (n = 3) |
| | SLTs (n = 4) |
| 4. Which solutions should we use to support self-management? | SSWA (n = 4) |
| | Family members (n = 2) |
| | SLTs (n = 4) |
| 5. Prototype solutions for supporting self-management | SSWA (n = 4) |
| | Family members (n = 2) |
| | SLTs (n = 4) |
| 6. How can we put the self-management approach we have developed in to practice? | SSWA (n = 5) |
| | Family members (n = 3) |
| | SLTs (n = 3) |

Abbreviations: SSWA-Stroke survivors with aphasia; SLTs- Speech and language therapists

the first meeting due to staff shortages in their service and one SLT was unable to attend the last meeting due to service pressures from COVID-19. No participants dropped out of the study.

## Intervention development

In this section we describe the process undertaken to develop the intervention. We have structured this section according to the stages outlined by the behaviour change wheel guide to designing interventions [28]. However, it is important to note that the discussion within the intervention development groups was more iterative. For example, in workshop three when generating 'solutions' for the intervention (stage two: identifying intervention options), it was natural for the discussion to progress to how the intervention might be delivered in practice (stage three: identifying interventions content and implementation options). We sought to encourage rather than constrain such discussion and participants naturally moved back and forth between creating a shared understanding of the behaviour, identifying options for the intervention and identifying how it could be implemented in practice. A summary of key outputs related to the stages of intervention development outlined by the BCW are shown in Fig 2.

**Stage one: Understand the behaviour.**  Prior to the intervention development groups and informed by the preparatory research, the research team undertook work to understand the behaviour (Stage 1a). Self-management is the overall target for the intervention, however, the research team did not define self-management as a single behaviour. Instead, the research team considered self-management as a 'system of behaviours' [28] as it does not occur in isolation but rather in the context of other behaviours (performed by the individual or performed by others) and which may interact with one another. Relevant behaviours were identified by reviewing findings from the systematic review work [6, 19] and qualitative fieldwork [35, 36] conducted previously (Stage 1b).

The findings of the preparatory research suggested that self-management is a highly context specific system of behaviours which may be influenced by a number of factors including the

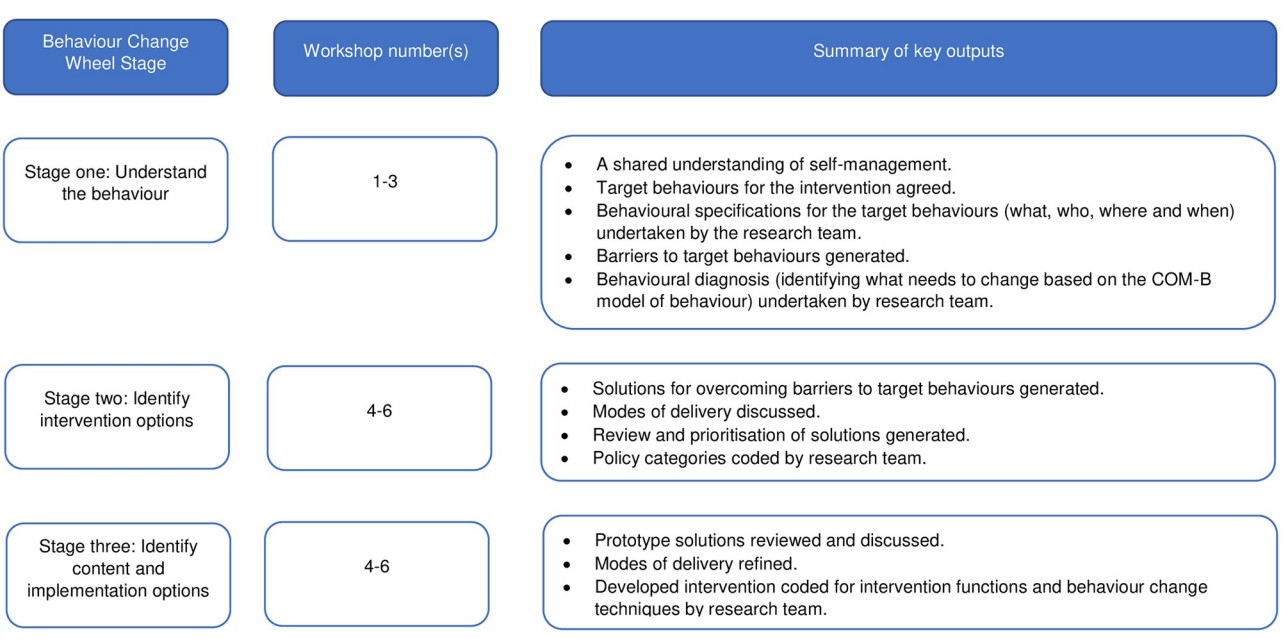

**Fig 2. Summary of key outputs linked to stages of intervention development.**

severity of the persons aphasia, their self-defined needs and wishes, and the availability of support. Thus, there was a tension between being specific in the behaviours to be targeted by the intervention but allowing the behaviours to be broad enough to be relevant to the population. Furthermore, Michie et al. [28] suggest limiting the number of behaviours to be targeted by the intervention. We generated a list of five candidate behaviours which included; 1. Communicating outside of the home, 2. Participating in meaningful activity, 3. Seeking and maintaining social support, 4. Obtaining information about stroke and aphasia and 5. Forming a partnership with healthcare providers. The list of candidate behaviours was not considered to be exhaustive. We undertook behavioural specifications (stage 1c) for each of the five candidate behaviours which included defining what the behaviour was, who was involved, where and when the behaviour occurred.

Although the research team undertook this preparatory work prior to the intervention development workshops, the target behaviours were not considered to be final. We wished to develop a shared understanding of self-management with the group and to prioritise which behaviours should be targeted as part of the intervention. The preparatory work was also used to guide our communication about self-management in the preparatory meeting.

In workshop one (preparatory meeting), the discussion on 'living well with aphasia' mainly focused on three topic areas: 1. Communicating with other people, e.g. being able to speaking to strangers, being understood, being given more time, other people having awareness of aphasia; 2. Doing things and having support, e.g. Getting out to places/enjoying them, going for a walk, having a support network, having support to pay bills or talk to companies; and 3. Feeling like you are living well, e.g. Having confidence to go out alone, laughing, not giving up. There was significant overlap between these discussions and the findings of the preparatory research.

During workshop two (talking about self-management), there was a general consensus to avoid the use of the term 'self-management'; participants felt the term lacked meaning and was too technical. The concept of 'living with aphasia' was preferred for describing the intervention to SSWA/family members. We retained 3 out of 5 of the pre-specified target behaviours (communicating outside of the home, seeking and maintaining social support, participating in meaningful activity) and refined the final two behaviours (obtaining information about stroke and aphasia, and forming a partnership with healthcare providers) in to one target behaviour (taking the lead in managing). This was based on discussions about needing information about self-management but also setting expectations about level of involvement within the intervention e.g. joint partnership which moves towards the SSWA and/or family taking the lead. The first author undertook a further behavioural specification for the behaviour of taking the lead in managing between workshops two and three.

In workshop three (what can we change to support self-management), participants discussed barriers ('what might stop me') to each of the target behaviours. Some example barriers were provided by the research team and additional barriers were generated by participants. After workshop three, the first author used information from the discussion to complete a behavioural diagnosis for each of the target behaviours (stage 1d). This was done by coding data using the COM-B model. For example, barriers to meaningful activity were coded to psychological capability (e.g. SSWA having an awareness of what opportunities might be in their local area such as aphasia groups, accessible exercise classes), social opportunity (e.g. such opportunities being communicatively accessible to SSWA) and reflective motivation (e.g. SSWA being confident to participate). For each of the target behaviours, potential targets for change were identified at the capability, opportunity and motivation level of the COM-B model.

**Stage two: Identifying intervention options.** During workshop three participants generated solutions to the barriers identified ('what could we do to help') (stage 2a) and discussed modes of delivery ('how could we make this happen') (Stage 3b). Example solutions to address

barriers to taking part in meaningful activities included; SLTs discussing and tailoring information about the opportunities in the local area (psychological capability), building up confidence in small stages (reflective motivation) and developing a routine (reflective motivation). There was overlap in discussion of modes of delivery across the target behaviours; in particular, participants agreed on the need for accessible written information for SSWA and information for family and friends to overcome barriers related to capability across more than one of the target behaviours e.g. knowing practical strategies to facilitate communication outside of the home or information about the joint role in therapy to facilitate taking the lead in managing. The solutions generated were reviewed and prioritised (including participants discussing whether there was anything missing) at the beginning of workshop four. We also asked participants to evaluate the strengths and weaknesses of existing written materials during workshop four to inform prototype intervention materials. Discussions included design e.g. minimising the amount of text, preferring a diversity of people to be represented in images, having a contents page, and on content e.g. liking the opportunity to personalise resources, the clear message that people with aphasia are not stupid, needing local information.

The policy category 'guidelines' (stage 2b) was identified to be of most relevance by the research team as the intervention is to be delivered by SLTs at the service level (agreed by participants in workshop three). The National Clinical Guideline for Stroke [15] recommends that self-management approaches are offered to all stroke survivors and this intervention seeks to develop a framework to support delivery of this for SSWA.

**Stage three: Identifying intervention content and implementation options.** Following workshop four, the research team developed a prototype intervention based on the prioritised solutions and participants preferences for the mode of delivery. The prototype intervention had four components including an accessible guide for SSWA, a guide for family and friends, training for SLTs and a toolkit for SLTs to implement the intervention in to practice. Participants agreed with the main components of the prototype intervention and there were no suggestions for additional components. Refinements to each of the components were suggested across workshop five and six (stage 2a; stage 3b). These included refinements to;

- Content: For example, additional information to be included in the guide for SSWA such as information about what a stroke/aphasia is and additional information in the training for SLTs about when to refer on for more specialist support (the boundaries of their role in supporting mental wellbeing within the intervention).

- Presentation of materials: For example, suggestions for additional illustrations within the guides and suggestions for where fewer illustrations may be useful. Suggested changes to wording e.g. family members preferring to be encouraged to be 'advocates' for people with aphasia rather than 'champions'.

- Mode of delivery: For example, clarification that the intervention should be 'integrated' within usual sessions rather than 'in addition' to usual sessions and the suggestion that worksheets which were included in the SLT toolkit should be held by SSWA and could be reviewed as part of planning for the end of therapy.

After the last workshop, the developed intervention was coded for intervention functions (stage 2a) and BCTs (stage 3a) (see Table 4 for examples). The coding was undertaken by the first author and checked by another researcher. Some components of the intervention aligned clearly with intervention functions and BCTs. For example, participants developed an idea for SSWA (and their families, where appropriate) to create a self-management plan for the end of therapy which aligns with BCT 1.4. Action planning. In some cases, the research team needed to further operationalise an idea which had been generated by participants. For example, SLTs

**Table 4. Examples from the behavioural analysis to map co-produced intervention strategies to the BCW.**

| Target behaviour | Barrier to the target behaviour | Co-produced intervention strategy | Target construct (COM-B)[a] | Intervention Function (BCW)[b] | BCTs[c] |
|---|---|---|---|---|---|
| **Communicating outside of the home** | Not having someone to practice with or provide practical support. | Support provided by SLT[d] or family/friends or SLT provides help to access locally available support e.g. from charities, communication partner schemes. | Opportunity | Enablement, environmental restructuring | 3.3. Social support (practical) 3.4. Social support (emotional) 7.1. Prompts/Cues 12.2. Restructuring the social environment |
| | SSWA[e] lacking confidence in their ability. | Having support to build up confidence gradually in small steps. | Motivation | Enablement, Persuasion | 4.1. Instruction on how to perform the behaviour 6.1. Demonstration of the behaviour 8.1. Behavioural practise or rehearsal 8.7. Graded tasks 2.7. Feedback on outcome of behaviour 8.3. Habit formation 10.4. Social reward |
| **Seeking or maintaining social support** | Family and friends not knowing what they can do to support the SSWA | Written, clear, easy read summary of what family and friends can do to help SSWA to communicate. Family and friends acting as advocates for the person with aphasia to make sure that other people in their social network are aware of what they can do to support them to communicate. | Capability Opportunity | Training Environmental restructuring | 4.1. Instruction on how to perform the behaviour 4.1. Instruction on how to perform the behaviour 13.1. Identification of self as a role model |
| **Participating in meaningful activity** | Not being able to do the same activities as before stroke | Being supported to develop a new routine (including awareness of locally available opportunities which are tailored to the SSWA's interests). | Capability Opportunity | Education Enablement Environmental restructuring | 4.1. Instruction on how to perform the behaviour 1.4. Action planning 2.3. Self-monitoring of behaviour 7.1. Prompts/cues 12.5. Adding objects to the environment |
| **Taking the lead in managing** | Thinking it is the health care professional's job to say what happens in therapy. | Increase knowledge about SSWA/family members having a joint role in therapy. | Capability | Education | 4.1. Instruction on how to perform the behaviour 5.1. Information about health consequences 5.6. Information about emotional consequences |

Abbreviations:

[a]COM-B- Capability Opportunity Motivation model of behaviour;

[b] BCW- Behaviour Change Wheel;

[c] BCTs- Behaviour Change Techniques;

[d] SLT-Speech and language therapist;

[e] SSWA- Stroke survivors with aphasia

felt that they needed training on interpersonal strategies they could use to support and encourage SSWA and their families to take the lead in managing. This was further operationalised to incorporate four BCTs in the skills training (4.1. Instruction on how to perform the behaviour, 6.1. Demonstration of the behaviour, 8.1. Behavioural practise or rehearsal and 2.2. Feedback

on behaviour). In these situations, the BCW framework acted as a useful checklist to maximise the inclusion of appropriate BCTs.

In total, the intervention targets six out of nine intervention functions including; education, persuasion, training, environmental restructuring, modelling and enablement. Twenty-nine BCTs are included from the BCT taxonomy [28] from the categories of goals and planning, feedback and monitoring, social support, shaping knowledge, natural consequences, comparison of behaviour, associations, repetition and substitution, comparison of outcomes, regulation, antecedents, identity, and self-belief. The developed intervention will be delivered on a one-to-one basis by trained SLTs working in community stroke services.

## Evaluation of co-production process

Participants gave constructive feedback at the end of each workshop which contributed to our facilitation of subsequent groups. In workshop one, we asked people to volunteer their reflections as they wished. However, one SSWA requested that we asked people to reflect one-by-one to make it easier for people with aphasia to contribute. The group agreed and we adopted this approach in subsequent workshops. One SSWA requested that we email the accessible summaries of the workshop discussion to them as they would like to share what they were doing in the groups with their family member. We asked the other participants if they would also like this and sent summaries as requested.

Overall, participants were positive about their attendance at the groups. SSWA, their family members and SLTs reported that they valued the mixing of smaller sub-groups for tasks (getting the chance to speak to all other participants). Two SSWA stated that it was helpful for them to be in a separate group from their family member and they had enjoyed the opportunity to be in a group just with other SSWA ('people who understand me'). Participants agreed about the lack of support for SSWA and valued the opportunity to contribute to research in this area. SSWA and their family members reported enjoying the peer support opportunity which was provided in the session. In the final evaluation, two SSWA said they would have liked more informal time for socialising (either at the beginning or end of the session). One SSWA and one family member said that they enjoyed the intellectual engagement provided during the workshop tasks. SLTs agreed that they had valued the groups and some reported that it had influenced their practice. One SLT reported it had made them think about preparing SSWA and their families for the longer-term. Another SLT reported using some of the terminology from the workshop (supporting 'living with' aphasia) and also reported that it had made them consider how they supported the families of people with aphasia in their service. SLTs thought the written materials and tools would be useful for their practice; they suggested there was a lack of materials and framework to support living with aphasia and that this might be a barrier to SLTs supporting self-management in practice. In the final evaluation of the workshops, SSWA and family members gave different opinions about the length of the workshop; one SSWA suggested they would have liked it to be shorter (at the end they felt fatigued) and one suggested that they liked the length of the group (as it gave them time to express themselves).

## Discussion

This paper outlines the process undertaken to develop a self-management intervention for SSWA. To our knowledge, this is the first self-management intervention to be developed specifically for this population. A large programme of work has been undertaken to develop the intervention. Our preparatory research included systematic reviews of existing literature [6, 19] and qualitative research [35, 36]. Following this, we ran a series of intervention

development groups with key stakeholders including SSWA, their family members and SLTs. Principles of co-production [26, 27] and an intervention development process incorporating behaviour change theory [28, 29] were used to inform the structure and content of the groups. In line with Medical Research Council guidance for the development of complex interventions [34], the developed intervention will be piloted and further refined in a feasibility study before a definitive evaluation.

## Benefits and challenges of using co-production

Co-production seeks to engage key stakeholders as partners so that the developed intervention is more likely to be feasible to deliver and acceptable to the people who will access it [32, 33]. To facilitate the co-production process, we aimed to create an environment in which participants were equal; where each participant's expertise was recognised and where participants felt empowered by seeing their contributions feed in to the development of the intervention [26, 39]. We used a number of strategies to incorporate principles of co-production into the intervention development groups. In particular, we dedicated part of the first workshop to introduce participants to the idea of co-production, reinforcing participants' expertise. To foster a sense of empowerment, we produced accessible summaries, which were reviewed at the beginning of each session to show how the group's ideas were contributing directly to the development of the intervention. The tasks were also designed to foster a sense of ownership, for example, by creating a shared understanding of the term self-management and agreeing upon how this term should be used in the intervention.

Aphasia presents a particular challenge to creating a sense of equivalence amongst participants in the co-production process [52, 53]. We were mindful of the potential power imbalance between SSWA and other participants and, as outlined in the method section, we used a number of strategies to maximise the accessibility of the workshops. To encourage collaboration, we utilised an ice breaker task where participants brought in a personal object to introduce themselves and so participants could get to know one another. We noted that this was a useful conversation starter and during the break, conversation about the objects continued. Another strategy which was particularly valued by SSWA was to mix the smaller sub-groups in which tasks were undertaken. Although, co-production focuses on collaboration of the group as a whole, SSWA valued having some discussions solely with other SSWA. This may be a useful strategy to overcome potential power imbalances; encouraging SSWA to share their experiences and reinforcing perceived ability to contribute.

Using strategies to maximise accessibility, we were able to successfully engage SSWA in co-production techniques which were both generative (generating ideas e.g. shared understanding of self-management, solutions for the intervention) and evaluative (e.g. evaluating prototype materials) [54]. A challenging task for all participants (including SSWA) was to generate solutions for the intervention as there seemed to be a natural drift towards focusing on barriers in the Discussion. The use of a local artist to facilitate discussion and draw solutions in real time was a successful approach which helped to engage participants and maximised accessibility for SSWA (who were able to work with the artist to create visual representations of their ideas). The artist had previous experience of working with stroke survivors.

## Benefits and challenges of using the BCW

The BCW [28] provided a useful framework for structuring the intervention development groups; offering a set of defined stages to focus the co-production tasks. However, like others [37], we found that some stages of the BCW translated more easily in to co-production activities than others. For example, for stage one (understand the behaviour), we were able to use

participants' views and experiences to refine the target behaviours and understand the barriers to these. On the other hand, for stage two (identifying intervention options) and stage three (identifying content and implementation options) we decided not to provide lists of intervention functions or BCTs for participants to consider. This was for two reasons; firstly, we felt it would have been difficult to do this in a way which was accessible to participants (in particular SSWA) and secondly, we did not wish to constrain participants' ideas or creativity in terms of generating solutions for the intervention [55]. Instead, we adapted the BCW approach by retrospectively coding intervention functions and BCTs to the strategies co-produced by participants.

## General challenges faced during the development process

Designing tasks for the intervention development groups required a significant investment of time. Although we were informed by existing studies and resources [53, 56, 57], each task (including materials and worksheets) was designed bespoke for this study. It was sometimes challenging to manage competing demands; firstly, in terms of the time constraints of the sessions, and ensuring that tasks were feasible to complete given the limited number of tasks which could be undertaken within a session; secondly, to ensure that tasks were engaging and accessible to all participants; and lastly, to ensure that the tasks were productive and aligned with the BCW framework.

## Strengths and limitations

In this article we transparently report the methods we used to develop a self-management intervention for SSWA. This study contributes to the complex intervention development literature by detailing how we integrated principles of co-production and behaviour change theory to develop an intervention. Such detailed reporting is necessary to increase our understanding of which methods lead to the development of effective interventions and whether an approach which integrates both a partnership and theory-based approach is beneficial [20]. Transparent reporting is particularly relevant for advancing complex intervention development in stroke, where intervention development methods are often poorly reported [25]. Furthermore, this study contributes to the literature by detailing the methods we used to include SSWA in the co-production process. There are currently limited studies in this area [53].

A strength of the study is the level of engagement from participants which is demonstrated by the high retention rate and positive feedback gained from participants during evaluations. Other studies have reported difficulties in engaging and retaining participants in co-production processes [58]. Due to unavoidable circumstances, some participants did not attend all sessions. We aimed to minimise the impact of this on the development process by including an introduction to each session (summarising of what had happened at the previous session), and, accessible written summaries of discussions. We acknowledge that the fieldnotes for the evaluation were taken and summarised by one researcher and that this is a limitation of the research.

A limitation is that our sample did not include participants with severe aphasia. Whilst we did not specifically exclude this group, participants did self-select and our recruitment strategy may have needed to be more targeted to include this population. We also recognise there would be significant challenges in including participants with severe aphasia in the co-production process; namely, in obtaining informed consent, in ensuring that participants with severe aphasia were able to keep pace with conversation amongst participants with less severe impairments, and, engage with abstract and complex constructs within the co-production tasks [52, 59, 60]. An alternative approach may be to include family members of people with severe

aphasia where it is not possible to obtain informed consent. Future research should consider how best to include participants with severe aphasia in the co-production process.

A further consideration with regards to sampling, is the extent to which a self-selected group of SSWA are representative of other SSWA. Those who self-select are likely to differ from the general population of SSWA and thus the developed intervention may be more acceptable to this group than the general population. However, we aimed to ensure a diverse range of perspectives were considered as far as possible in the design of the intervention; a) by including evidence from our preparatory research and b) by including SLTs in the process so that their experience of working with a range of SSWA could contribute to the development process.

A limitation of this article is that we have chosen not to fully describe the intervention we developed. We plan to evaluate the intervention in a cluster randomised controlled trial (RCT) and we wish to avoid the potential for contamination between intervention and control sites. We plan to fully describe the intervention according to reporting guidelines [61] in future publications.

## Implications for self-management

Self-management interventions are well established in other long-term conditions [62], however, have only been developed and tested in stroke more recently [12]. There is still uncertainty about when such approaches should be delivered in the stroke pathway, in what format and by whom [12]. Based on their synthesis of literature across long-term conditions, Taylor et al. [62] advocate providing individually tailored and condition specific self-management support. The findings of this study support this; the strategies which were co-produced often highlighted self-management needs which arose directly as a result of aphasia e.g. limited access to accessible written information to support self-management, building confidence in communication, family members knowing strategies to help SSWA to communicate. This suggests that there may be value in developing a self-management intervention to address the specific needs of SSWA.

Existing stroke self-management interventions vary in their theoretical underpinnings [19] although are often associated with self-efficacy theory [63]. In contrast to previous interventions [19], our development process is based on a framework which draws together cumulative knowledge from a range of behaviour change theories [28, 29]. This may be advantageous in ensuring a wide range of theories are considered in the design of the intervention and in specifying how intervention components are thought to facilitate change. Understanding which components of self-management interventions are effective is vital for providing specific recommendations about how self-management should be incorporated in the stroke pathway [12].

## Conclusions

This article provides a detailed description of the development of a self-management intervention for SSWA using co-production and behaviour change theory. By combining two approaches, we aimed to harness the strengths of each approach. In particular, through co-production, increasing the likelihood of developing an intervention which is feasible and acceptable in practice. And through behaviour change theory (BCW), defining the active components of the intervention which will help us to refine the intervention and develop a reasoned account of why it is or is not effective in the future. Combining both approaches was resource intensive, however, through this experience, this study provides other researchers with ideas and strategies for how this may be done in practice. We also add to the

literature in terms of examples of how we maximised accessibility for SSWA involved in the co-production process. The developed intervention is, to our knowledge, the first to be developed specifically to address barriers to self-management posed by aphasia. The developed intervention will be subject to further evaluation and refinement in a planned feasibility study. A definitive cluster RCT is planned in the future.

## Supporting information

**S1 File. Example worksheet.**
(PDF)

**S2 File. Example infographic.**
(PDF)

**S3 File. Example of artist illustrations generated during workshop three.**
(PDF)

**S4 File. Example of accessible summary.**
(PDF)

## Acknowledgments

We wish to express our thanks to the stroke survivors with aphasia, family members and speech and language therapists who participated in the study for their time and valuable insights. Thanks also to Dr Jessica Hall for co-facilitating the co-production workshops and to Dr Ian Kellar for checking coding to the behaviour change wheel. We also wish to thank Dr Jennifer Hall for providing advice about linking co-production activities to the BCW and the Aphasia team (including Dr Abi Roper and Dr Lucy Dipper) at City University London for their advice and suggestions about designing co-production activities for people with aphasia. Finally, thank-you to artist Tom Bailey for helping to facilitate workshops three and four.

## Author Contributions

**Conceptualization:** Faye Wray, David Clarke, Madeline Cruice, Anne Forster.

**Data curation:** Faye Wray.

**Formal analysis:** Faye Wray.

**Funding acquisition:** Faye Wray, David Clarke, Madeline Cruice, Anne Forster.

**Methodology:** Faye Wray, David Clarke, Madeline Cruice, Anne Forster.

**Project administration:** Faye Wray.

**Supervision:** David Clarke, Madeline Cruice, Anne Forster.

**Writing – original draft:** Faye Wray.

**Writing – review & editing:** Faye Wray, David Clarke, Madeline Cruice, Anne Forster.

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
