## [Decision Letter · Decision Letter 0]

23 Jun 2021

PONE-D-21-11777

Development of a self-management intervention for stroke survivors with aphasia using co-production and behaviour change theory

PLOS ONE

Dear Dr. Wray,

Thank you for submitting your manuscript to PLOS ONE. After careful consideration, we feel that it has merit but does not fully meet PLOS ONE’s publication criteria as it currently stands. Therefore, we invite you to submit a revised version of the manuscript that addresses the points raised during the review process.

Please can you consider the reviewer's comments and provide a response to each comment made. 

We look forward to receiving your revised manuscript.

Kind regards,

Andrew Soundy

Academic Editor

PLOS ONE

“The authors wish to thank The Stroke Association for providing the funding for this study.”

“Funding for this study was provided by a postdoctoral fellowship awarded to FW by The Stroke Association (Ref: SA PDF 19\\100011) (https://www.stroke.org.uk/research/support-programme-aphasia-life-after-stroke). Authors DC, MC and AF were co-applicants on this fellowship. The funder did not play a role in the design of the study, or in the collection and analysis of data, or in writing the manuscript, or in the decision to publish."

Additional Editor Comments (if provided):

Reviewers' comments:

Reviewer's Responses to Questions

**Comments to the Author**

1. Is the manuscript technically sound, and do the data support the conclusions?

Reviewer #1: Yes

Reviewer #2: Yes

2. Has the statistical analysis been performed appropriately and rigorously? 

Reviewer #1: N/A

Reviewer #2: N/A

3. Have the authors made all data underlying the findings in their manuscript fully available?

Reviewer #1: No

Reviewer #2: Yes

4. Is the manuscript presented in an intelligible fashion and written in standard English?

Reviewer #1: Yes

Reviewer #2: Yes

5. Review Comments to the Author

Reviewer #1: Major:

1. I would recommend to use the COREQ and SRQR (https://www.equator-network.org/reporting-guidelines) for reporting qualitative studies. Although there are items in these checklists, which are not relevant for this study, it can be used. Especially, the method section could be improved by providing more details on selection of the participants and data analyses.

2. The discussion section could be improved by comparing the findings of current study to other research and theories. In addition, a broader implications of the research should be explained as well.

Minor and specific:

Title

The title could be approved as follows: ‘’How to develop a complex self-care management intervention for aphasia for stroke survivors? A process description using co-production and behavior change theory’’.

Abstract

It is concise and clear, all relevant topics are addressed.

Background

Explain why partner-based and theory-based approach was chosen over other categories mentioned by O’Cathain et al.

The research question/aim should be stated at the end of the background section.

Discuss and explain: are there any comparable studies done using the same approach in different disease areas for the development of self-management interventions? If yes, what kind of development process did they follow?

Methods

Data availability; please explain and add information relevant to this topic.

Describe COAST and FAST tests in more details (lines 183-184)

Explain: ‘’NHS band 6 or above’’ (line 178)

For the following four topics, check reporting guidelines as well.

Regarding the topic ‘’Personal characteristics’’

• Add section on this and describe all relevant characteristics of the authors or workshop leaders

Regarding the topic ‘’Participant selection’’: add more information related to this topic, e.g.

• How were participants selected?

• How many people refused to participate or dropped out? Reasons?

• Why did you not tried to include family members of patients with severe aphasia in the study as participation of this group of patients was too difficult?

Regarding the topic ‘’Analyses of data’’

• How were the field notes analyzed?

• How was coding done? By means of software or only manually?

Regarding ‘’data collection‘’

• Was data saturation a goal?

• Recording? Face to face? Online workshop?

Results

Were all family members relatives of the included patients?

Section intervention development (page 15 (row 341) to page 20 (row 503); although it contains a lot of relevant information; it is very long; try to give a brief summary of the different development stages and put the other details in an appendix. A figure containing a summary what was done in each step would be helpful for the reader.

A brief overview of the intended intervention should be presented. e.g. details on de modes of delivery and the description of different intervention components should be added.

Content table 3 is only explained very brief. E.g. indicate limited number of family members present in half of the workshops, give a reason for being absent.

Explain abbreviations in tables 2 and 4

Explain briefly COM-B model using a footnote (page 18, line 426)

Discussion

The sections discussing observations of current research could be grouped together in order to provide a clear overview of recommendations for future researchers which want to develop complex interventions in general and specific for aphasia (lines 558 to607, 623 to 630)

Discuss the impact of selection of a specific group of stakeholders for the generalizability of the intervention for all aphasia patients.

The analysis of the field notes was done primarily by one author; discuss this limitation in the discussion section.

Discuss whether the participation of not all stakeholders in every workshop may have influenced development process?

Conclusion

No comments

Reviewer #2: The paper is well-written and deeply detailed. Readers, even from research different areas, are able to replicate the process used to co-produce the intervention. There are also a lot of examples useful to better understand the process. However, some parts are a bit heavy and not easy to follow, for this reason, I have some suggestions:

In the method section: add a diagram describing all the steps done (from the preparatory research to the development of the intervention) otherwise the reader get lost

In the method section: the paragraph "Integration of evidence from preparatory research" appear disconnected from the discourse, maybe it could be enclosed previously in the first method paragraph "Preparatory research"

The self-management implications derived from the intervention development are not so clearly stated, maybe it would be better to stress this more in the discussion and conclusion sections.

6. PLOS authors have the option to publish the peer review history of their article (what does this mean?). If published, this will include your full peer review and any attached files.

Reviewer #1: **Yes: **Ghislaine van Mastrigt, MSc, PhD, Maastricht University, The Netherlands

Reviewer #2: No

---

## [Author Response · Author response to Decision Letter 0]

19 Aug 2021

As instructed in the decision email, responses to specific reviewer and editor comments are provided in the 'response to reviewers' file which has been uploaded.

---

## [Decision Letter · Decision Letter 1]

13 Oct 2021

Development of a self-management intervention for stroke survivors with aphasia using co-production and behaviour change theory: An outline of methods and processes.

PONE-D-21-11777R1

Dear Dr. Wray,

We’re pleased to inform you that your manuscript has been judged scientifically suitable for publication and will be formally accepted for publication once it meets all outstanding technical requirements.

Kind regards,

Andrew Soundy

Academic Editor

PLOS ONE

Additional Editor Comments (optional):

Reviewers' comments:

Reviewer's Responses to Questions

**Comments to the Author**

1. If the authors have adequately addressed your comments raised in a previous round of review and you feel that this manuscript is now acceptable for publication, you may indicate that here to bypass the “Comments to the Author” section, enter your conflict of interest statement in the “Confidential to Editor” section, and submit your "Accept" recommendation.

Reviewer #1: All comments have been addressed

Reviewer #2: All comments have been addressed

2. Is the manuscript technically sound, and do the data support the conclusions?

Reviewer #1: Yes

Reviewer #2: Yes

3. Has the statistical analysis been performed appropriately and rigorously? 

Reviewer #1: N/A

Reviewer #2: Yes

4. Have the authors made all data underlying the findings in their manuscript fully available?

Reviewer #1: Yes

Reviewer #2: Yes

5. Is the manuscript presented in an intelligible fashion and written in standard English?

Reviewer #1: Yes

Reviewer #2: Yes

6. Review Comments to the Author

Reviewer #1: Dear Authors,

I am very pleased with the way you have answered my questions and the comments. In addition, you have been able to incorporated them in the revised version of the paper.

I do not have any other suggestions for improvement,

kind regards,

reviewer 1

Reviewer #2: (No Response)

7. PLOS authors have the option to publish the peer review history of their article (what does this mean?). If published, this will include your full peer review and any attached files.

Reviewer #1: **Yes: **Ghislaine A.P.G. van Mastrigt, PhD, Maastricht University, The Netherlands

Reviewer #2: No

---

## [Editor Report · Acceptance letter]

15 Nov 2021

PONE-D-21-11777R1 

Development of a self-management intervention for stroke survivors with aphasia using co-production and behaviour change theory: An outline of methods and processes 

Dear Dr. Forster:

I'm pleased to inform you that your manuscript has been deemed suitable for publication in PLOS ONE. Congratulations! Your manuscript is now with our production department. 

Kind regards, 

on behalf of

Dr. Andrew Soundy 

Academic Editor

PLOS ONE